# Neonatal Early Onset Sepsis (EOS) Calculator plus Universal Serial Physical Examination (SPE): A Prospective Two-Step Implementation of a Neonatal EOS Prevention Protocol for Reduction of Sepsis Workup and Antibiotic Treatment

**DOI:** 10.3390/antibiotics11081089

**Published:** 2022-08-11

**Authors:** Francesco Cavigioli, Francesca Viaroli, Irene Daniele, Michela Paroli, Luigi Guglielmetti, Elena Esposito, Francesco Cerritelli, Gianvincenzo Zuccotti, Gianluca Lista

**Affiliations:** 1Neonatology and Neonatal Intensive Care Unit, V. Buzzi Children’s Hospital, 20100 Milan, Italy; 2Department of Biomedical and Clinical Science, Università Degli Studi di Milano, 20100 Milan, Italy; 3Non-Profit Foundation C.O.ME. Collaboration, Centre for Osteopathic Medicine Collaboration, 65121 Pescara, Italy

**Keywords:** early onset sepsis, newborns, antibiotics, mortality, infection, serial clinical examination

## Abstract

Current neonatal early-onset sepsis (EOS) guidelines lack consensus. Recent studies suggest three different options for EOS risk assessment among infants born ≥35 wks gestational age (GA), leading to different behaviors in the sepsis workup and antibiotic administration. A broad disparity in clinical practice is found in Neonatal Units, with a large number of non-infected newborns evaluated and treated for EOS. Broad spectrum antibiotics in early life may induce different short- and long-term adverse effects, longer hospitalization, and early mother-child separation. In this single-center prospective study, a total of 3002 neonates born in three periods between 2016 and 2020 were studied, and three different workup algorithms were compared: the first one was based on the categorical risk assessment; the second one was based on a Serial Physical Examination (SPE) strategy for infants with EOS risk factors; the third one associated an informatic tool (Neonatal EOS calculator) with a universal extension of the SPE strategy. The main objective of this study was to reduce the number of neonatal sepsis workups and the rate of antibiotic administration and favor rooming-in and mother–infant bonding without increasing the risk of sepsis and mortality. The combined strategy of universal SPE with the EOS Calculator showed a significant reduction of laboratory tests (from 33% to 6.6%; *p* < 0.01) and antibiotic treatments (from 8.5% to 1.4%; *p* < 0.01) in term and near-term newborns. EOS and mortality did not change significantly during the study period.

## 1. Introduction

Neonatal sepsis represents an important problem among the neonatal population. It is a systemic infection occurring in infants at <28 days of life, defined as early or late depending on the age of onset. EOS is defined as a culture-proven bloodstream infection or bacterial meningitis occurring within 7 days of a newborn’s life with clinical signs and symptoms consistent with sepsis [1,2,3,4].

The occurrence of morbidity and mortality of EOS has declined in the last three decades, mainly due to worldwide screening panels for prenatal infectious diseases, as well as enhanced maternal and neonatal protocols for EOS risk management. The Centers for Disease Control and Prevention (CDC) estimated that incidence has reduced from more than 1.5/1000 births in the early 1990s to less than 0.5/1000 births in recent years. Despite this positive trend, EOS still represents a serious problem for the neonatal population [5,6].

Risk factors for early-onset neonatal sepsis include maternal, perinatal, and neonatal factors [1]. Well-known organisms causing early-onset neonatal sepsis are typically colonizers of the maternal genitourinary tract. These can include Group B Streptococcus (GBS) and *Escherichia coli*, which play a major role in the etiology of this disease [4,7].

Symptoms of EOS are often non-specific; most of the time, they can appear at birth or in the first hours of life, depending on the involved bacteria and on the severity of the infection, but the clinical manifestation of an infection can also appear with a delay of 48–72 h from birth, potentially rapidly evolving into dramatic diseases [2,3].

The standard diagnosis of sepsis remains a challenge: it requires a positive blood culture, whereas laboratory tests have shown poor predictive value [6,8,9,10]. Despite this, laboratory tests, and in particular C-reactive protein (CRP), are still widely used as a decision-making tool for the diagnosis and treatment of EOS [11,12]. Instead, assessment of perinatal risk factors is a key point in deciding if intrapartum antibiotic prophylaxis (IAP) is needed for the mother to prevent neonatal EOS, but it is well known that severe EOS can still occur in newborn infants from mothers without any perinatal risk factors.

American Academy of Pediatrics (AAP) released recent guidelines suggesting three different approaches for the prevention of EOS among newborns at ≥35 Weeks GA [13]:The categorical risk assessment: in this approach, pre- and perinatal risk factors are used to guide the administration of Intrapartum Antibiotic Prophylaxis (IAP) to the mother and the consequent management of newborns at risk for EOS.The multivariate risk assessment (the Neonatal Early-Onset Sepsis Calculator) is a predictive model based on information available in the immediate perinatal period. This model establishes a prior probability for newborn sepsis, which could be combined with neonatal physical examination [14,15,16].Risk assessment based on newborn clinical condition: serial physical examination (SPE), regardless of maternal risk factors, is performed at standardized time points by caregivers who detail clinical observations at standard intervals. Infants are then treated empirically with antibiotics if they appear ill at birth or develop signs of illness over the first 48 h after birth [8].

Each of these strategies has some limitations: the categorical risk assessment is associated with a higher number of non-infected newborns evaluated and treated for EOS compared with the EOS Calculator and the SPE approach. Concerning the EOS calculator, even though it recommends routine clinical care for low-risk infants, the absence of perinatal risk factors cannot exclude the potential for sepsis. Moreover, the presence of mild symptoms (mainly respiratory maladaptation) could induce the caregiver to early unnecessary interventions [17].

Clinical signs are an early marker of EOS, showing more sensitivity when compared with laboratory tests [8,9,18], but the use of the SPE alone raises concern about the real possibility of carrying out adequate specific training for the personnel involved in the care of the newborn in different hospital contexts [17].

The purpose of this study was to investigate if the combination of the latter two validated strategies could reduce their individual limitations. The main goal was to significantly decrease the rate of sepsis workups and antibiotic treatments without increasing the risk of sepsis and mortality.

## 2. Results

This study enrolled a total of 3002 newborn infants ≥35 weeks GA born consecutively at the same institution in Milan (Italy) throughout three 4-month periods between November 2016 and March 2020. They were admitted to the NICU or Special Care Unit only when antibiotic treatment was needed. Neonates with major malformations, and those requiring antibiotic prophylaxis for surgical diseases or other invasive procedures, were excluded from the study.

Each population consisted of approximately 1000 infants. Weight, GA, APGAR score, gender, positive maternal GBS status (including positive rectovaginal swab or genitourinary GBS infection during pregnancy), and percentage of meconium-stained amniotic fluid at birth were comparable among the three populations (Table 1). A comparison of the three populations for other perinatal risk factors reveals statistically significant differences in ROM, maternal fever, and mode of delivery.

The key results are summarized in Table 2 and Table 3.

The number of sepsis workups gradually decreased from 33% of the population in P1 to 11.4% in P2 and 6.6% in P3 (*p* < 0.01). Compared to P1, the odds ratio (OR) of sepsis workup decreased significantly in P2 (OR 0.26, CI 0.20–0.33) and even more in P3 (OR 0.14, CI 0.11–0.19).

The empiric antibiotic treatment rates were 8.5% in P1 and 7.3% in P2, significantly decreasing to 1.4% in P3 (*p* < 0.01). Among the empirically treated infants in P1, only 5/85 showed clinical symptoms at the beginning of therapy (5.8%). This percentage increased in P2 to 9.5% (7/73 infants) and 71.4% in P3 (10/14 infants). Among the treated infants, culture-proven sepsis was found in 1/85 in P1 (1.1%), 3/73 in P2 (4.1%), and 3/14 in P3 (21.4%). The late discontinuation (>96 h) of therapy was frequent in P1 (54/85 = 63%). In P2, this number decreased to 5/73 (6.8%). In P3, 9/14 received prolonged antibiotic treatment (64%).

Compared to P1, the OR of antibiotic treatment was similar in P2 (OR 0.9, CI 0.65–1.25) but decreased significantly in P3 (OR 0.16, CI 0.09–0.27).

The multivariate logistic analysis showed an increased likelihood of antibiotic administration associated with maternal fever during labor and PROM. In fact, logistic regression demonstrated that maternal fever increased the administration of antibiotic treatment 77 times (OR 77, CI 44.3–139) and PROM > 48 h 35 times (OR 35, CI 15.9–74.6).

The length of hospitalization increased in P2 compared to P1 (4.16 days versus 3.20 days) but decreased significantly in P3 (3.03 days versus 3.20 days, *p* = 0.04).

In the three populations, the number of culture-proven EOS was 1/999, 3/1003, and 3/1000, respectively (0.1%, 0.3%, and 0.3%), with no statistically significant difference (*p* = 0.29). The clinical characteristics and EOS risk factors of newborns with culture-proven sepsis in the three populations are synthesized in Table 4.

## 3. Discussion

Several approaches have been suggested for the management of newborns at-risk for EOS, but the recent literature highlighted some limitations that are consistent with the results we found in this study. Combining the EOS Calculator and the Universal SPE approach was effective at minimizing some of the risks without increasing any adverse effects.

The overall incidence of culture-proven EOS in the population studied (7/3002 = 2.3/1000 live births) is presumably overestimated because of a limited study period. To our knowledge, EOS incidence is variable in the term and near-term newborn population between 0.5 and 1/1000 LB and did not differ from these data in our center in the last years.

In P1, according to the in-use protocol, we performed laboratory tests and administered antibiotic therapy to the newborns subsequently diagnosed with EOS because of risk factors such as prematurity and PROM.

Notably, in P2, one of the three infants who developed sepsis had no pre- or perinatal risk factors. He underwent routine neonatal care but, at 14 h of life, during rooming-in, presented with a dramatic onset of respiratory and circulatory symptoms, rapidly developing into meningitis and death. Blood and cerebrospinal fluid cultures were positive for GBS. This patient would not have been diagnosed earlier either with the P1 protocol (because of the absence of risk factors) or with the EOS calculator alone, which would have assigned a low risk score at birth. Only the application of a universal SPE would have likely intercepted the onset of the first symptoms earlier.

In P3, two EOS cases were diagnosed by means of the SPE form since newborns were asymptomatic with a low EOS risk score at birth. The third case showed clearer risk factors (maternal fever and unknown GBS status), he was symptomatic at birth, and his blood culture resulted rapidly positive for *E. coli*. This patient would have benefited from any of the three protocols.

The categorical risk assessment, as the one applied in the first population (P1) of our study, is associated with a large number of infants undergoing laboratory evaluation [19]. Although numerous laboratory exams have been recommended over time for the diagnosis of EOS, they have shown to possess poor predictive values [20]: serial low WBC count and low absolute neutrophil count are associated with increased odds of infection, but normal values of these parameters do not possess enough sensitivity to rule out EOS in neonates [20,21,22]. CRP is not an effective selective EOS tool because it shows low sensitivity during the early phases of the disease and very low specificity. This can lead to misdiagnosis and subsequent antibiotics overtreatment because elevations of this protein may also occur in non-infectious conditions (meconium aspiration syndrome, surfactant application, maternal fever at delivery, high birth weight, perinatal asphyxia, or prolonged rupture of membranes) [23,24]. In the first population in our study, a high percentage of newborns (63% of treated infants) received prolonged antibiotic courses due to elevated values of CRP, but sepsis was ruled out in 98.8% of them. Other laboratory tests like procalcitonin (PCT) or presepsin are gaining interest in recent literature, but additional studies are needed before they can be widely adopted into the EOS management algorithm [25,26,27]. Peripheral blood culture remains the gold standard for the diagnosis of sepsis. Recent studies have evaluated umbilical cord blood culture as a possible tool for EOS diagnosis, even though this is still not validated [28].

Unnecessary laboratory tests and unneeded antibiotic administration show short- and long-term consequences: infant discomfort and parental stress, admissions to NICU and, therefore, the interruption of parental bonding and breastfeeding, prolonged length of hospital stay, and increased health care costs [19,29,30]. Early neonatal antibiotic exposure has been recently demonstrated to be correlated to asthma, allergic and autoimmune disease, microbiome alteration, and obesity [31,32,33,34,35,36]. Strategies avoiding unnecessary pre- and postnatal antibiotics and favoring mother-newborn bonding and breastfeeding should be implemented [34]. The restriction of the criteria to perform blood tests and the elimination of CRP tests from the laboratory panels in P2 brought a drastic reduction in venipuncture, even though the protocol was still based on a categorical risk assessment. The discontinuation of antibiotic treatment with the confirmation of negative blood cultures in this population likely allowed fewer patients to be treated for shorter amounts of time. Moreover, in P2, an SPE form was introduced for the population at risk of EOS to potentially reduce antibiotics administration even further, given that clinical signs indeed increase the accuracy of EOS diagnosis [8,9,18,37]. Despite the earlier discontinuation of antibiotics, the length of hospitalization in P2 appeared to be higher than in P1, presumably because of a progressive but cautious introduction and application of the SPE form by the caregivers.

Limiting the clinical examination only to newborns with EOS risk factors can lead to missed diagnoses in newborns with no perinatal warnings [38,39,40]. We report the case of one patient of the P2 population, with no pre- or perinatal risk factors, who developed a severe form of GBS sepsis that led to his death. GBS colonization status remains one of the most important metrics to evaluate, but this information can be inaccurate at the time of birth. Our patient who died from fulminant GBS early-onset sepsis was born to a mother whose vaginal swab tested negative at 35 wks of gestation. Several studies have found that a GBS PCR assay is more feasible and offers a better predictive value of GBS EOS than the rectovaginal swab culture [41,42,43], but more studies are needed before wide implementation. After this adverse event, we analyzed the recently introduced EOS protocol to verify its safety compared to the previous one, concluding that even applying the old protocol, the newborn would have died.

The “universal SPE approach” suggested by the GBS Prevention Working Group of Emilia-Romagna puts the clinical status of newborns in the foreground and even includes newborns with no risk factors. The clinical presentation of early-onset sepsis can be delayed, mild, and non-specific, though it can rapidly evolve into dramatic diseases. Extending the SPE approach to the entire newborn population can reduce missed diagnoses. Berardi et al. showed that the universal SPE approach could produce a sharp reduction in laboratory exams and antibiotic treatments without a consequent increase in EOS [34,44].

This approach does not consider any of the prenatal risk factors and requires extensive experience and training from health professionals. The learning curve before the safe utilization of the SPE form is something to be taken into account. The timing of examinations to record clinical signs are the same in risk and non-risk infants. To overcome concerns in recognizing early and mild signs of sepsis due to individual variability of clinical assessment and differences in skills, we decided to combine universal SPE with the EOS calculator.

The EOS calculator is a validated computer tool freely available on the internet [15]. The EOS risk assigned by the calculator at birth, which is strongly dependent on pre- and perinatal maternal risk factors (e.g., GBS status, maternal intrapartum antibiotic therapy and intrapartum prophylaxis, highest intrapartum maternal temperature, gestational age, duration of ruptured membranes), is combined with the clinical status of the newborn (e.g., well-appearing, equivocal, or clinical illness) as a crucial factor for the diagnosis of suspected EOS. The combination of different risks and clinical statuses correspond to different workup and treatment protocols [14,15,16]. Recent publications demonstrated that the EOS calculator, compared to conventional management strategies, shows lower relative risks for empirical antibiotic therapy without affecting safety [16,45,46].

Though promising, the EOS calculator still has some limitations. Based on the initial assessment at birth, it assigns the category of routine neonatal care to a substantial percentage of EOS cases, which may lack any initial pathological clinical signs of EOS [38,47,48]. Furthermore, it may not distinguish symptoms that may be the consequence of a different pathological state [38]. For example, transient tachypnea of the newborn presents itself with a high respiratory rate and oxygen requirement soon after birth, but following the EOS calculator classification may suggest performing laboratory tests and antibiotics administration. In contrast, following the observation-based approach, infants can be further monitored before interventions. Symptoms such as skin discoloration (pallor, jaundice, or cyanosis) or prolonged capillary filling, as well as other symptoms related to circulation problems, are of great importance, but these are not part of the EOS computer test.

A combined strategy, including the neonatal EOS calculator and the universal SPE approach, allowed our institution to overcome the limitations of the two single approaches. The drastic decrease in the antibiotic treatment rate was likely mainly due to the introduction of the EOS calculator, but the extensive use of the SPE form in the entire population has resulted in a further reduction. In particular, when applying the serial physical examination, great importance was placed on skin discoloration and circulatory signs, whereas an attitude of prudent clinical waiting for early mild respiratory symptoms was allowed.

## 4. Materials and Methods

The study was conducted between November 2016 and March 2020 at the Ospedale dei Bambini “V. Buzzi”, Milan, Italy, delivering about 3200 newborns per year, starting from a two-step revision of our local protocol for prevention of EOS approved by the Hospital Committee (REF: IDF_BUZ_TIN_29 Rev.00 18.11.2019). Data regarding EOS laboratory tests, antibiotic treatment, and clinical outcomes were collected on three subsequent populations of about 1000 consecutive infants of ≥35 wks GA, born at our institution, and were the subject of the three-degree thesis. Consent for clinical data collection in routine clinical settings was signed by parents at birth in our institution.

Population 1 (P1; n = 999) was studied between 1 November 2016, and 28 February 2017. In this period, the approach to preventing EOS in our institution was based on categorical risk assessments. The presence of risk factors was, per se, an indication for the performance of newborn laboratory tests: repeated complete blood count (CBC), CRP, and blood culture tests were performed, and antibiotic treatment was started in asymptomatic infants when any of the three exams were positive or in the presence of clinical symptoms for sepsis. Antibiotic treatment was withdrawn only when CRP returned to normal values.

As our rate of EOS workup and antibiotic treatment was higher than the data reported in the literature, we decided to revise our protocol according to recent guidelines.

Population 2 (P2; n = 1003) was studied between 1 January and 5 May 2018, after the first implementation of the protocol. The main changes that we introduced consisted of a restriction of the indications for EOS workup, the elimination of CRP testing from the EOS workup scheme, and the introduction of a previously published Serial Physical Examination (SPE) form for those infants with maternal or perinatal EOS risk factors [8]. Careful clinical examinations of asymptomatic but at-risk newborns were applied at regular time frames during the entire hospital stay, aiming to detect early clinical signs of infection to guide the neonatologist’s decisions on whether to continue the clinical surveillance or perform laboratory tests and administer antibiotics. When antibiotics were started, CRP testing was performed to guide the duration of treatment. In the presence of negative blood culture and in absence of clinical symptoms, antibiotic treatment was withdrawn.

Population 3 (P3; n = 1000) was studied from 2 December 2019, to 31 March 2020, as a result of the second step of our protocol implementation. We introduced an online tool, the EOS calculator, in the routine care of all newborns ≥35 weeks GA and we extended the SPE form to the entire population as a universal tool to detect early signs of disease in newborns with no maternal or perinatal EOS risk factors.

The combination of these two validated strategies was standardized in the protocol. The main deviations from the strict EOS calculator indications consisted in the application of the Serial Physical examination also to infants classified by the EOS Calculator as needing routine neonatal care and in allowing the caregivers to retard the application of laboratory tests and empirical antibiotic treatment in infants showing only mild respiratory symptoms (as possible signs of retarded respiratory transition) in the absence of other systemic signs. As in P2, withdrawal of empiric antibiotic treatment was decided in the presence of negative blood culture, repeated negative CRP, and the quick resolution of clinical symptoms.

### Statistical Analysis

Absolute frequencies (percentages) for qualitative data and means (standard deviations) for quantitative data were used to describe the demographic characteristics of the sample. Then, the differences between the three time periods were evaluated using 2 tests for qualitative data and analysis of variance (ANOVA) with one between-factor for quantitative data.

The search for and verification of missing data were performed using a variety of techniques [49].

Comparing the three study periods yielded odds ratios (ORs) for the dichotomous outcomes, i.e., the likelihood of administering antibiotics treatment and the likelihood of performing exams on neonates.

Regarding the continuous outcomes, an ANOVA with one between-factor (the time period) was conducted, followed by Tukey’s post hoc test.

As an exploratory analysis, the unadjusted and adjusted ORs between antibiotics administration and neonatal exams and the following variables, i.e., time period, sex, GA, labor type, birthweight, PROM, TV, maternal fever, and amniotic fluid, were calculated using univariate and multivariable logistic regressions, respectively.

Two multivariate logistic regressions were performed as exploratory analyses of continuous outcomes: for the length of stay, the predictors were time period, sex, GA, labor type, birthweight, PROM, TV, maternal fever, and amniotic liquid; for the duration, in days, of antibiotic treatment, the predictors were time period, sex, GA, labor type, birthweight, PROM, TV, maternal fever, and amniotic liquid.

Alpha = 0.05 was defined as the level of statistical significance for two-sided tests. The statistical analysis was conducted using the R program (R Core Team, R Foundation for Statistical Computing, version 3.6.2, Vienna, Austria).

## 5. Conclusions

To our knowledge, this is the first time that a combined strategy of universal SPE and EOS calculator has been suggested. In our prospective observational study, this approach is associated with a significant reduction in laboratory testing and antibiotic prescriptions. Our combined strategy is not linked to an increased incidence of neonatal sepsis and mortality. The possible positive consequences of these results are reduced neonatal discomfort, increased rooming-in, breastfeeding and mother–infant bonding, less parental anxiety, and the reduction of hospitalization length with the possible reduction of economic burden on public health. The short- and long-term consequences of early antibiotics administration could also be potentially reduced.

## Figures and Tables

**Table 1 antibiotics-11-01089-t001:** Characteristics of the populations.

	P1	P2	P3	*p*-Value
N	999	1003	1000	
GA, wks (mean)	39 + 4	39 + 4	39 + 4	0.45 *
Mean weight, g (SD)	3292 (461)	3320 (463)	3340 (453)	0.06 *
Rectovaginal swabs performed, n (%)	705 (76.8%)	847 (85%)	835 (84%)	0.48 ^§^
GBS-positive status, n (%)	114 (16%)	112 (13%)	126 (15%)	0.56 ^§^
GBS status unknown, n (%)	205 (20%)	154 (15.3%)	161 (16.1%)	0.43 ^§^
PROM > 18 h, n (%)	99 (10.7%)	178 (18%)	208 (21%)	0.01 ^§^
Maternal fever > 38 °C, n (%)	30 (3.3%)	53 (5.2%)	69 (6.9%)	0.01 ^§^
Mode of delivery-vaginal, n (%)	761 (76.2%)	798 (79.5%)	811 (81.1%)	<0.01 ^§^
Meconium-stained amniotic fluid, n (%)	134 (13.4%)	155 (15.5%)	139 (13.9%)	0.43 ^§^

Absolute frequencies (percentage) for qualitative data and mean (standard deviation) for quantitative data were used to describe the demographic characteristics of the sample. *p*-values: * ANOVA; ^§^ Chi-square.

**Table 2 antibiotics-11-01089-t002:** Main outcomes.

	P1	P2	P3	*p*-Value
N	999	1003	1000	
Sepsis Laboratory test, n (%)	332 (33%)	115 (11.4%)	66 (6.6%)	<0.01 ^§^
Antibiotic treatment, n (%)	85 (8.5%)	73 (7.3%)	14 (1.4%)	<0.01 ^§^
Culture-proven EOS, n (%)	1 (0.1%)	3 (0.3%)	3 (0.3%)	0.29 ^§^
Days of Hospitalization, mean days (SD)	3.20 (1.5)	4.16 (1.44)	3.03 (1.73)	0.04 *

Absolute frequencies (percentage) for qualitative data were used to describe the main results. *p*-values: ^§^ Chi-square; * statistically significant difference from the comparison between P1 and P3 with Tukey posthoc analysis.

**Table 3 antibiotics-11-01089-t003:** Odds ratio among the study populations regarding clinical outcomes.

	P2 vs. P1	P3 vs. P1	P3 vs. P2
Antibiotic treatment	0.90 (0.65–1.25)	0.16 (0.09–0.27)	0.18 (0.1–0.31)
Neonates exam	0.26 (0.20–0.33)	0.14 (0.11–0.19)	0.55 (0.4–0.75)

Comparing the three study periods yielded odds ratios (ORs) for the dichotomous outcome.

**Table 4 antibiotics-11-01089-t004:** Clinical characteristics and EOS risk factors of newborns with culture-proven sepsis.

N	Study Period	GA	GBS Status	Other Risk Factors	IAP	Onset of Symptoms	Lab Test Prior to Symptoms	Antibiotics Prior to Symptoms	SPE	EOS Score at Birth	Etiology
1	P1	36 + 5	neg	no	no	5 h	yes	no	no	n.a.	GBS
2	P2	39 + 2	neg	no	no	14 h	no	no	no	n.a.	GBS
3	P2	39 + 6	neg	PROM	no	12 h	no	no	yes	n.a.	*E. coli*
4	P2	40 + 4	neg	Chorio-amnionitis	2 doses	No symptoms	yes	yes	yes	n.a.	GBS
5	P3	38 + 4	unknown	Maternal fever	no	At birth	no	no	yes	0.62	*E. coli*
6	P3	36 + 2	neg	PROM	2 doses	9 h	no	no	yes	0.41	*E. coli*
7	P3	40 + 2	neg	PROM	no	48 h	no	no	yes	0.3	GBS

## Data Availability

The data presented in this study are available on request from the corresponding author.

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
