# Peer review of "Neonatal Early Onset Sepsis (EOS) Calculator plus Universal Serial Physical Examination (SPE): A Prospective Two-Step Implementation of a Neonatal EOS Prevention Protocol for Reduction of Sepsis Workup and Antibiotic Treatment"

_antibiotics, 2022, doi:10.3390/antibiotics11081089_

Round 1

Reviewer 1 Report

I have a few comments for the authors that I hope will add to the quality of the Manuscript

Exclude referencing to Figure from abstract, also please write all abbreviations in full when first mentioning them

Figure 1, poor resolution, superscripts need explanation in footnote

Reference 1 can be mentioned only at the end of the sentence. Put reference 4 before the period.

I am not sure this 90ties is the correct spelling, please revise.

Table 1 and table 2 should include references

Figure 2 has poor resolution, also superscript symbols should be explained in the footnote

you can write this >=35 as ≥35

Please include ethics approval information in the material and methods section

Table 3, last row, there is a degrees symbol C that is not needed?

Data are missing from tables, you write only whole numbers, while on the side you put (%), percentages should be written in brackets next to whole numbers

Statistical tests used should be mentioned in the footnotes, also data presentation

This sentence needs to be rewrite to clarify what exactly is it you wanted to say g. In fact, in the three populations, maternal fever is associated with an odds ratio (OR) of 77 (CI 44,3-139) and PROM > 48h is associated with an odds ratio (OR) of 35 (CI 15,9-74.6) of neonatal antibiotic treatment.

Line 149 - missing reference

Reference 49 needs to be aligned, references should be excluded from conclusion

please consider writing discussions starting with your results, simply discussing everything else is not a quality discussion, also everything mentioned in the conclusion section should be previously discussed.

Author Response

Thanks to reviewer 1 for comments.

Referencing to figure was excluded from abstract. Figure 1, Figures 2 and 3 and Tables 1 and 2, were eliminated indeed, the reader can look this information directly on publications referenced. Ref 1 was mentioned at the end of the sentence. We put ref 4 before the period. 90ties was corrected, thankyou. We included references of Hospital committee approval of the protocol for management of newborns at risk for EOS. 

In table 1,2,3 the degree symbol was corrected, percentages have been written in brackets next to whole numbers. Statistical tests used and data presentation have been mentioned in the footnotes

Sentence of ORs has been clarified

Line 149 was erased as the discussion needed to be shortened. Reference 49 aligned, references excluded from conclusions. Discussion was renewed starting with results, shortening the literature review considerations.

English language was revised in total with the help of a native English speaking person.

Reviewer 2 Report

In this paper, Dr. Cavigioli and co-authors proposed a new EOS diagnosis protocol using a combined strategy of universal SPE and EOS calculator, which showed drastic reduction in antibiotic administration and OR of antibiotic treatment without increasing the risk of sepsis. This study is very interesting and could be helpful for future EOS prevention protocol development. However, I have a few suggestions and comments which I think should be addressed before being published, as listed below:

1.      Introduction could be improved.

a.      The introduction elaborated on the limitations of laboratory tests for sepsis diagnosis and current guidelines for EOS preventions but didn’t cover the limitation of EOS calculator and why it alone is not sufficient for EOS prevention. I think adding this part in the introduction offers smoother transition into study design and result section and give people a clearer picture of why the combined strategy will be superior.

b.      At the end of the discussion, neither did the authors summarize experiment design and goals, nor did they mention how this study would improve upon existing protocols and how the combined strategy should be adopted in future EOS protocols. Because of these gaps, I couldn’t see the importance of this study.

c.      Page 5 Line 103~106 should be revised.

2.      Results could be clearer.

a.      Table 3 should add OR comparison of the 3 population, the result should also mention the false positive and false negative rate of sepsis workups and antibiotic treatment.

b.      The potential reasons why P2 had longer days of hospitalization as comparted to the P1 & P3 should be addressed.

3.      Discussion could be improved.

a.      The discussion should elaborate on the main outcome of the study and how universal SPE approach complements neonatal EOS calculator.

Author Response

Comments and Suggestions for Authors

In this paper, Dr. Cavigioli and co-authors proposed a new EOS diagnosis protocol using a combined strategy of universal SPE and EOS calculator, which showed drastic reduction in antibiotic administration and OR of antibiotic treatment without increasing the risk of sepsis. This study is very interesting and could be helpful for future EOS prevention protocol development. However, I have a few suggestions and comments which I think should be addressed before being published, as listed below:

  1. Introduction could be improved.
  2. The introduction elaborated on the limitations of laboratory tests for sepsis diagnosis and current guidelines for EOS preventions but didn’t cover the limitation of EOS calculator and why it alone is not sufficient for EOS prevention. I think adding this part in the introduction offers smoother transition into study design and result section and give people a clearer picture of why the combined strategy will be superior.

Thanks for your comments. We elaborated the introduction with EOS calculator and SPE limitations.

  1. At the end of the discussion, neither did the authors summarize experiment design and goals, nor did they mention how this study would improve upon existing protocols and how the combined strategy should be adopted in future EOS protocols. Because of these gaps, I couldn’t see the importance of this study.

The reviewer probably intended at the end of the introduction. At the end of introduction we summarized the purpose of the study. In methods we summarized the study design and described how the two strategies can combine each other.

  1. Page 5 Line 103~106 should be revised.

This section was shortened

  1. Results could be clearer.
  2. Table 3 should add OR comparison of the 3 population, the result should also mention the false positive and false negative rate of sepsis workups and antibiotic treatment.

We added a table were ORs comparing the three population are added for antibiotics and laboratory tests. We added in the results a description of number of infants evaluated and treated and how many of them were symptomatic or asymptomatic and how many were true septic.

  1. The potential reasons why P2 had longer days of hospitalization as comparted to the P1 & P3 should be addressed.

It is not clear but we provided an hypothesis.

  1. Discussion could be improved.
  2. The discussion should elaborate on the main outcome of the study and how universal SPE approach complements neonatal EOS calculator.

Was done.

Reviewer 3 Report

This single center study aims to evaluate whether the neonatal sepsis risk calculator can be combined with serial clinical observation Several studies scattered around the world now compare approaches, and data from these comparisons are therefore welcome. However, the authors need to remove some repetitive or uninteresting parts and spend more text to clarify aspects that remain uncertain in this version. It is not clear how these 2 approaches (sepsis risk calculator and enhanced observation) are combined; I suggest a more detailed description of them. For example, when applying the sepsis calculator, some asymptomatic infants or infants with mild symptoms in the first hours of life reach a score that suggests performing examinations and administering empirical antibiotics, whereas with the observation-based approach, infants can be monitored without giving empirical antibiotics. How are the contrasts between the 2 approaches resolved?

which one of the 2 approaches clinicians decided to follow? The text has some inaccuracies. English needs a substantial revision.

Specific comments:

Abstract

the abstract is very discursive and gets lost in the description of the different approaches (it almost reads like an introduction), but it does not provide the aims of the study and numerical data on the results of the study (what is really of interest to the reader). How many infants in the study? over what period? what are the statistically significant results and conclusions?

Furthermore, unusual for an abstract, there is a reference to a figure (Figure 1). It should be removed

Main text

Introduction:

Page 2-6, lines 34-115: the introduction is disproportionately long and should be greatly shortened, summarizing the suggested approaches in a few lines (20-30 lines ?). Similarly, Figures 2 and 3 and Tables 1 and 2 are unnecessary, the reader can look this information directly at the officila recommendations of the American Academy of Pediatric or the Centers for Disease Control and Prevention.

In addition, usually the purposes of the study are briefly described at the end of the introduction; they should be added

Methods:

This section is too long and statistical analyses can be summarized. Tab 5 should probably be removed and the content summarized in a few lines.

For clarity, methods can be presented by summarizing them in short paragraphs (e.g., Period 1, from xxx to xxx, the approach was based on xxx; Period 2, from xxx to xxx, the approach was based on xxx etc.).

Results

It would be very interesting to know how many asymptomatic or symptomatic neonates were treated with empirical antibiotics according to the 3 strategies adopted; and how many of them (symptomatic or asymptomatic) underwent sepsis workup. Among neonates treated with antibiotics, how many had culture-proven sepsis or symptoms prompted a diagnosis of culture-negative sepsis?

It is not clear from the text or tables whether the rates of infants treated with empirical antibiotics refer only to infants admitted to the NICU or include also neonates admitted to the special care. Also, compared with all previous studies that used the sepsis risk calculator, the rates of infants treated in period 3 are really very low, and it is not clear whether investigators included in the calculation also neonates with major malformations, those with surgical prophylaxis, or those asphyxiated, etc.. Please clarify

Page 7, line 138-139: "...the incidence of EOS was 1, 3, and 3, respectively...", probably the authors mean 1/1000 live births; However, the table 4 shows EOS % 0.1, 0.1 and 0.3. It is unclear if the authors refer to culture-proven os suspected EOS. Please clarify

Discussion:

Page 7-8, Line 143-208: too long. It seems a review (on PCR, PCT, Presepsin and WBC) rather than a research article. Summarize in a couple of sentences

Page 9-10, Line 208-275:

Tables:

Table 3 and 4, please add full numbers (and % or SD in brackets); a p value calculated only on % rates does not have sense

Table 3, Meconium stained amniotic fluid …0438°C ?? Please, correct

Table 4, last line: “P1-P3”, this information (the comparisons you presumably made) should be given in the manuscript or in the title of the table, not at the bottom

Author Response

This single center study aims to evaluate whether the neonatal sepsis risk calculator can be combined with serial clinical observation Several studies scattered around the world now compare approaches, and data from these comparisons are therefore welcome. However, the authors need to remove some repetitive or uninteresting parts and spend more text to clarify aspects that remain uncertain in this version. It is not clear how these 2 approaches (sepsis risk calculator and enhanced observation) are combined; I suggest a more detailed description of them. For example, when applying the sepsis calculator, some asymptomatic infants or infants with mild symptoms in the first hours of life reach a score that suggests performing examinations and administering empirical antibiotics, whereas with the observation-based approach, infants can be monitored without giving empirical antibiotics. How are the contrasts between the 2 approaches resolved?

which one of the 2 approaches clinicians decided to follow? The text has some inaccuracies. English needs a substantial revision.

Thanks for your Comments. The text has been shortened and some repetitive parts were eliminated. In the Methods we tried to better describe the three approaches used in the three populations and mainly the third approach in particular how the EOS calculator can combine with the SPE approach. English was revised.

 Specific comments:

Abstract

the abstract is very discursive and gets lost in the description of the different approaches (it almost reads like an introduction), but it does not provide the aims of the study and numerical data on the results of the study (what is really of interest to the reader). How many infants in the study? over what period? what are the statistically significant results and conclusions?

Furthermore, unusual for an abstract, there is a reference to a figure (Figure 1). It should be removed

Was revised as suggested

Main text

Introduction:

Page 2-6, lines 34-115: the introduction is disproportionately long and should be greatly shortened, summarizing the suggested approaches in a few lines (20-30 lines ?). Similarly, Figures 2 and 3 and Tables 1 and 2 are unnecessary, the reader can look this information directly at the officila recommendations of the American Academy of Pediatric or the Centers for Disease Control and Prevention.

In addition, usually the purposes of the study are briefly described at the end of the introduction; they should be added

Was revised as suggested

Methods:

This section is too long and statistical analyses can be summarized. Tab 5 should probably be removed and the content summarized in a few lines.

For clarity, methods can be presented by summarizing them in short paragraphs (e.g., Period 1, from xxx to xxx, the approach was based on xxx; Period 2, from xxx to xxx, the approach was based on xxx etc.).

Was revised as suggested

Results

It would be very interesting to know how many asymptomatic or symptomatic neonates were treated with empirical antibiotics according to the 3 strategies adopted; and how many of them (symptomatic or asymptomatic) underwent sepsis workup. Among neonates treated with antibiotics, how many had culture-proven sepsis or symptoms prompted a diagnosis of culture-negative sepsis?

In Results we added data on symptoms and Antibiotic treatment. As numbers are low we just put a description and didn't get an analysis of that.

It is not clear from the text or tables whether the rates of infants treated with empirical antibiotics refer only to infants admitted to the NICU or include also neonates admitted to the special care.

Clarified.

Also, compared with all previous studies that used the sepsis risk calculator, the rates of infants treated in period 3 are really very low, and it is not clear whether investigators included in the calculation also neonates with major malformations, those with surgical prophylaxis, or those asphyxiated, etc.. Please clarify

Clarified.

Page 7, line 138-139: "...the incidence of EOS was 1, 3, and 3, respectively...", probably the authors mean 1/1000 live births; However, the table 4 shows EOS % 0.1, 0.1 and 0.3. It is unclear if the authors refer to culture-proven os suspected EOS. Please clarify.

Clarified.

Discussion:

Page 7-8, Line 143-208: too long. It seems a review (on PCR, PCT, Presepsin and WBC) rather than a research article. Summarize in a couple of sentences

Was revised as suggested

Page 9-10, Line 208-275:

Tables:

Table 3 and 4, please add full numbers (and % or SD in brackets); a p value calculated only on % rates does not have sense

Was done

Table 3, Meconium stained amniotic fluid …0438°C ?? Please, correct

Was done

Table 4, last line: “P1-P3”, this information (the comparisons you presumably made) should be given in the manuscript or in the title of the table, not at the bottom

Was done

Round 2

Reviewer 3 Report

the manuscript is greatly improved. I have only a few additional comments.

- Despite the small cohort, the number of culture proven EOS is very high (7 cases among 3002 neonates, that is 2.3/1000 live births). Can the authors comment on this finding?

- I suggets adding a table summarizing the 7 cases of EOS (clinical diagnosis, hours of life at the onset of symptoms, IAP exposure, sepsis risk calculator score, etc) to clarify with which of the different approaches the diagnosis was timely

-  It is certainly safer to extend clinical observation to all not-at-risk neonates. However, it involves relevant caregiving efforts. Did the authors use the same number and timing of visits as neonates with risk factors?

- how do the authors explain the large reduction of antibiotic use in period 3? by the introduction of the sepsis risk calculator? or through deviations (which they say they do) from what it suggests?

Author Response

Comments and Suggestions for Authors

the manuscript is greatly improved. I have only a few additional comments.

- Despite the small cohort, the number of culture proven EOS is very high (7 cases among 3002 neonates, that is 2.3/1000 live births). Can the authors comment on this finding? Data regarding the populations studied have been collected during a four months period each. We can presume that the incidence of EOS is overestimated because of this limited period. To our knowledge, EOS incidence is variable in the term and near term newborns population between 0,5 and 1/1000 LB and did not differ from these data in our center in the last few years.

- I suggets adding a table summarizing the 7 cases of EOS (clinical diagnosis, hours of life at the onset of symptoms, IAP exposure, sepsis risk calculator score, etc) to clarify with which of the different approaches the diagnosis was timely.

We added a table 4 with clinical characteristics of the 7 infected infants. We also added a paragraph in the discussion that could be redundant, so feel free to ask to eliminate if the reviewer think that the table is enough to clarify the different approaches.

-  It is certainly safer to extend clinical observation to all not-at-risk neonates. However, it involves relevant caregiving efforts. Did the authors use the same number and timing of visits as neonates with risk factors? Yes. After an extensive and careful training to caregivers involved in the newborn care, all newborns were examined in the same way. Was clarified in the discussion.

- how do the authors explain the large reduction of antibiotic use in period 3? by the introduction of the sepsis risk calculator? or through deviations (which they say they do) from what it suggests? A paragraph was added with our possible explanation at the end of the discussion.

In the file that we uploaded with revisions requested, we did not remove any of the sentences or paragraphs that was previously contained in the text. The sentences that were added were highlighted in yellow color.